# Neuro-Ophthalmologic Variability in Presentation of Genetically Confirmed Wolfram Syndrome: A Case Series and Review

**DOI:** 10.3390/brainsci13071030

**Published:** 2023-07-05

**Authors:** Ruben Jauregui, Nicolas J. Abreu, Shani Golan, Joseph F. Panarelli, Meenakshi Sigireddi, Gopi K. Nayak, Doria M. Gold, Janet C. Rucker, Steven L. Galetta, Scott N. Grossman

**Affiliations:** 1Department of Neurology, New York University Grossman School of Medicine, New York, NY 10016, USA; ruben.jauregui@nyulangone.org (R.J.); nicolas.abreu@nyulangone.org (N.J.A.); doria.gold@nyulangone.org (D.M.G.); janet.rucker@nyulangone.org (J.C.R.); scott.grossman@nyulangone.org (S.N.G.); 2Department of Ophthalmology, New York University Grossman School of Medicine, New York, NY 10016, USA; shani.golan@nyulangone.org (S.G.); joseph.panarelli@nyulangone.org (J.F.P.); 3Department of Medicine, New York University Grossman School of Medicine, New York, NY 10016, USA; meenakshi.sigireddi@nyulangone.org; 4Department of Pediatrics, New York University Grossman School of Medicine, New York, NY 10016, USA; 5Department of Radiology, New York University Grossman School of Medicine, New York, NY 10016, USA; gopi.nayak@nyulangone.org

**Keywords:** optic atrophy, neuro-ophthalmology, diabetes mellitus, neurogenetics, genotype–phenotype correlation, autosomal recessive, Wolfram syndrome

## Abstract

Wolfram syndrome is a neurodegenerative disorder caused by pathogenic variants in the genes *WFS1* or *CISD2*. Clinically, the classic phenotype is composed of optic atrophy, diabetes mellitus type 1, diabetes insipidus, and deafness. Wolfram syndrome, however, is phenotypically heterogenous with variable clinical manifestations and age of onset. We describe four cases of genetically confirmed Wolfram syndrome with variable presentations, including acute-on-chronic vision loss, dyschromatopsia, and tonic pupils. All patients had optic atrophy, only three had diabetes, and none exhibited the classic Wolfram phenotype. MRI revealed a varying degree of the classical features associated with the syndrome, including optic nerve, cerebellar, and brainstem atrophy. The cohort’s genotype and presentation supported the reported phenotype–genotype correlations for Wolfram, where missense variants lead to milder, later-onset presentation of the Wolfram syndrome spectrum. When early onset optic atrophy and/or diabetes mellitus are present in a patient, a diagnosis of Wolfram syndrome should be considered, as early diagnosis is crucial for the appropriate referrals and management of the associated conditions. Nevertheless, the condition should also be considered in otherwise unexplained, later-onset optic atrophy, given the phenotypic spectrum.

## 1. Introduction

Wolfram syndrome (WS) is typically an autosomal recessive disorder caused by biallelic variants in *WFS1* or *CISD2.* The gene *WFS1* is located on chromosome 4p16 and consists of eight exons [1,2,3]. *WFS1* encodes for the protein wolframin, an 890-amino acid transmembrane protein with nine segments located primarily in the endoplasmic reticulum [2,3]. Given that the endoplasmic reticulum plays an important role in protein folding, pathogenic variants in *WFS1* cause an accumulation of misfolded proteins that ultimately cause cell apoptosis [2]. The majority of reported pathogenic variants are in exon 8 of *WFS1*, which encodes the transmembrane and C-terminal domain of wolframin, which is important for its function [2]. Wolframin is ubiquitously expressed, and varying levels of expression in different tissues account for the clinical manifestations of this disorder, with high expression observed in the brain and pancreatic β-cells [1,3,4]. 

The diagnostic criteria for WS are met clinically by the presence of early onset insulin-dependent diabetes mellitus (DM) and bilateral optic atrophy (OA) [3,5]. Wolfram-associated DM differs from type 1 DM in that its onset is earlier and destruction of pancreatic β-cells is mediated by endoplasmic reticulum stress, rather than an autoimmune process [6]. Diabetes insipidus (DI) and sensorineural deafness (D) complete the classical tetrad of “DIDMOAD,” but the complete phenotype is only seen in approximately 50% of patients [3]. DM is typically the first manifestation of the syndrome, presenting in the first decade of life, followed by bilateral OA early in the second decade [7]. Around 65% of patients develop sensorineural deafness and 70% develop central DI, both typically observed in the second decade of life. Additional neuropsychiatric symptoms and urinary tract abnormalities can also develop later in life [3,5]. In this study, we describe a series of cases that illustrates the variable clinical presentations of WS in a tertiary neuro-ophthalmology referral clinic.

## 2. Cases

### 2.1. Patient 1

A 6-year-old girl with a history of insulin-dependent DM presented to the neuro-ophthalmology clinic for evaluation of slowly progressive bilateral vision loss. The patient reported to her parents that she had multiple episodes of transient vision loss on a background of persistent progressive loss. On examination, visual acuities were 20/50 OD and 20/70 OS on the near card. Pupils were large and minimally reactive with no afferent pupillary defect observed. Fundoscopy revealed bilateral generalized disc pallor with sharp borders (Figure 1). She perceived the control Ishihara plate OD and 1/12 plate OS. Visual fields were preserved with respect to counting fingers in all quadrants and her efferent exam was normal, including full ductions in all gaze directions. The rest of her neurologic exam was unremarkable. MRI brain/orbits with gadolinium contrast were unrevealing. Optical coherence tomography (OCT) revealed thinning of the retinal nerve fiber layer (RNFL) to 37 μm and 38 μm on the right and left eye, respectively. Laboratory work-up was normal, including vitamin B12, folate, and thiamine, heavy metal panel, treponemal antibodies, rapid plasma reagin (RPR), anti-nuclear (ANA), myelin oligodendrocyte (MOG), and aquaporin-4 (AQP-4) antibodies. Genetic testing revealed two truncating variants in trans in the gene *WFS1*, c.334C>T:p.(Q112*) and c.958_961delinsTCC:p.(P320Sfs*39) (Figure 2).

### 2.2. Patient 2

A 32-year-old man was referred for evaluation of dyschromatopsia and inability “to see any colors”. He reported noticing worsening color vision five years ago, with more pronounced deficits in the last two years. Significant medical history included antibody-negative DM, diagnosed at the age of 27. On exam, visual acuity was 20/20 OU at distance. He perceived 1/12 Ishihara plates bilaterally. Pupils were symmetric, sluggishly reactive to light, and had no afferent pupillary defects. Fundoscopy revealed temporal optic disc pallor bilaterally (Figure 3A,B). Ocular ductions were full and the rest of the neurologic exam unrevealing. MRI brain/orbits with gadolinium revealed mild thinning of the optic nerves bilaterally. OCT RNFL revealed thinning to 54 μm and 52 μm on the right and left eye, respectively. Humphrey visual field (HVF) testing did not reveal clear pattern of visual field loss. Laboratory work-up was unrevealing, including toxic (copper, heavy metal panel), metabolic (levels of Vitamins B1, B3, B6, B9, B12, E), and infectious (RPR, quantiferon gold) etiologies. Genetic testing revealed the homozygous variant *WFS1* c.1672C>T:p.R558C.

### 2.3. Patient 3

A 46-year-old woman presented for assessment of acute visual decompensation superimposed on chronic visual loss. She had a medical history of bilateral optic atrophy with poor vision for many years and carried a diagnosis of color blindness since childhood. Family history was significant for a sister with similar poor vision and DM. On examination, visual acuity was 20/50 bilaterally. She perceived only the control Ishihara plate with each eye. Pupils were reactive without an afferent defect. Fundoscopy revealed pale optic discs bilaterally (Figure 3C,D). Ocular ductions were full and the rest of the neurologic exam was unrevealing. MRI brain/orbits with gadolinium revealed bilateral symmetric optic nerve atrophy and OCT RNFL revealed thinning to 46 μm bilaterally. HVFs showed an enlarged blind spot on the right, and a dense inferior nasal defect on the left (Figure 3E,F). Laboratory work-ups, including Vitamin B12 levels, RPR, ANA, were all unrevealing. Genetic testing revealed the variants c.2254G>T:p.E752* and c.1673G>A:p.R558H in the gene *WFS1*.

### 2.4. Patient 4

A 45-year-old woman was referred to the neuro-ophthalmology clinic for evaluation of chronic optic atrophy with progressive vision decline over the preceding year and bilateral tonic pupils. Medical history was significant for neurogenic bladder, DM with onset at age 9, and OA of unknown etiology since her twenties. On examination, visual acuity was 20/40 OD and 20/70 OS. She did not perceive the control Ishihara plate on either eye. Pupils were dilated bilaterally, non-reactive to light, and pupillary ruff was absent on slit lamp examination. Fundoscopy revealed generalized optic disc pallor bilaterally, more severe temporally where the disc appeared excavated (Figure 4A,B). Ocular ductions were full in range and mild right-beating nystagmus was observed. The rest of her neurological exam was notable for no appendicular ataxia on finger-to-nose testing but mildly unstable gait and inability to walk in tandem. HVFs revealed bilateral temporal field defects, more pronounced in the left eye (Figure 4C,D). OCT RNFL revealed bilateral thinning to 47 μm on both eyes. The instillation of dilute pilocarpine (0.1%) to each eye constricted both pupils, confirming the diagnosis of tonic pupils (Figure 3E,F). MRI brain/orbits with gadolinium revealed diffuse atrophy of the optic nerves, chiasm, and tracts, without a compressive lesion, along with diffuse brainstem and cerebellar atrophy (Figure 5). Laboratory testing was unrevealing, including vitamin B12, treponemal testing, AQP-4, and MOG antibodies. Genetic testing revealed the variants c.1230_1233del:p.V412Sfs*29 and c.1672C>T:p.R558C in *WFS1.*

## 3. Discussion

This case series describes the heterogeneous clinical presentation of four genetically confirmed WS patients (Table 1). In our cohort, we observed WS presenting with symptoms of progressive vision loss (Patients 1, 3, and 4), color vision changes (Patient 2), and pupillary dilation (Patient 4). All patients had bilateral optic atrophy, with either retained (Patient 2) or decreased visual acuity (Patients 1, 3, and 4). In addition, all patients had markedly impaired color vision, even the patient with preserved acuity. Decreased visual acuity and color vision changes secondary to optic atrophy are common in WS [8]. In a study of 18 patients between the age of 5 and 15, decreased visual acuity was observed in 85% of patients, color vision defects in 94%, visual field defects in 100%, and optic disc pallor in 94% of patients [8]. Other reported ophthalmologic manifestations include nystagmus, strabismus, reduced corneal sensitivity, ophthalmoplegia, cataracts, and pigmentary retinopathy [8,9,10,11,12]. Interestingly, Patient 4 presented with tonic pupils, which, although atypical, has been reported in WS [13]. In a study of nine WS patients, three were described as having dilated pupils that reacted poorly to light and normally to accommodation, with subsequent pupillary constriction when dilute cholinergic drops were instilled [13]. Patient 4 also had a history of neurogenic bladder, which is commonly seen in WS, as an estimated 60–90% of patients develop urinary tract abnormalities around the third decade of life [7]. The presence of tonic pupils and neurogenic bladder can be secondary to autonomic dysfunction, which is commonly present in patients with diabetes [13]. In our cohort, none of the patients presented with the full WS tetrad, observed in around 50% of WS patients in total. All of our patients presented with OA [3]. This may reflect referral bias to a neuro-ophthalmology clinic. Despite DM typically presenting before OA, Patient 3 did not report a history of DM. Although rare, patients with genetically confirmed WS presenting with OA without DM have been reported [3,5,14]. In a study of 59 patients, there were three patients who did not present with DM, one of them presumed to be because of young age, while the other two were adults [5]. Although the sister was not examined in our clinic, Patient 3 reported having a sister with OA and DM, which suggests a role for epigenetic factors in the development of the phenotype spectrum of WS. 

Various characteristic, though non-specific, signs on MRI have been reported in WS patients. These include atrophy of the optic nerves, cerebellum, or brainstem, absence of the posterior pituitary bright spot, and T1 hypointensity/T2 hyperintensity in the pons [15,16,17]. From our cohort, Patients 3 and 4 had optic nerve atrophy, Patient 2 had mild optic nerve atrophy, while only Patient 4 also exhibited cerebellum and brainstem atrophy, along with T2 hyperintensities in the pons (Figure 4). These changes observed in MRI are secondary to the neurodegenerative nature of the disease, with a median age of death reported at 30 years, usually from respiratory failure secondary to brainstem atrophy [7].

The differential diagnosis for etiologies that cause optic disc pallor can be characterized based on laterality, acuity, and age group. Acute or subacute monocular lesions are most commonly caused by ischemic optic neuropathies, such as arteritic/non-arteritic anterior ischemic optic neuropathy or posterior ischemic optic neuropathy, which in general tend to be seen in patients over the age of 50 [18,19]. Inflammatory causes, such as optic neuritis, would cause acute vision loss, but episodes are commonly unilateral and optic disc atrophy is seen months after the initial insult, whereas the disc would appear hyperemic in the acute setting [20]. Other etiologies to consider include infiltrative, compressive, genetic, and toxic optic neuropathies, which are less common and typically present with slow, painless, bilateral disc atrophy. Infiltration of the optic nerve can be observed secondary to neoplastic processes, such as in lymphoma or leukemia, or inflammatory cells, such as in sarcoidosis, tuberculosis, or syphilis [18]. Toxic optic neuropathies can result from nutritional deficiencies, such as vitamin B12, medications such as ethambutol, or toxins such as methanol [18,21]. Genetic etiologies include Leber hereditary optic neuropathy, dominant optic atrophy, and various other systemic genetic disorders [22,23]

Multiple studies have elucidated genotype–phenotype correlations in WS, which are an important tool for the clinician as they help predict the age of onset and severity of the different clinical features. In an analysis of 96 patients, it was determined that the presence of two inactivating variants (nonsense or frameshift) predisposed to an earlier age of onset for both DM and OA [24]. Age of onset for both DM and OA occurred later when the patients harbored one or two missense variants [5]. Late-onset WS, when patients develop OA and DM at age 15 or later, has been almost exclusively associated with missense variants [25]. Additionally, a study of WS in a Japanese population reported that patients with predicted loss-of-function variants had earlier age of onset for OA and DM, while a different study reported that the degree of wolframin expression was correlated to the degree of visual impairment from WS-associated OA [26,27]. 

The patients in our series support the above genotype–phenotype correlations. From our cohort, Patient 2 had the mildest presentation of WS, as he developed OA and DM only in his late twenties. He was homozygous for the p.R558C missense variant, which is found in 1.34% of the Ashkenazi Jewish population and has been previously reported as causing a milder, later-onset variant of WS [28]. In a study of eight homozygotes, DM was diagnosed on average at the age of 19 and OA at the age of 29 [28]. A different study of eight homozygotes reported an average age of 18 years for the diagnosis of DM, while OA was observed in only one patient [29]. Although Patient 4 in our cohort was heterozygous for the p.R558C variant, she had earlier symptom onset compared to Patient 2. Given the reported genotype–phenotype correlations, this is likely because her second *WFS1* variant was frameshift. The most severe presentation of WS in our cohort was in Patient 1, as this patient had OA and DM at the age of 6. DM is typically the first manifestation of the syndrome, presenting in the first decade of life, followed by bilateral OA early during the second decade of life [3,7]. We speculate that the early symptom onset in Patient 1 is likely explained by her genotype, as she carried two inactivating variants, one nonsense and another frameshift. The p.Q112* variant observed in Patient 1 has been reported in the literature in a homozygous patient [30]. As expected given the two nonsense mutations, that patient was reported to have developed DM and OA at an early age, at 5 and 11 years, respectively, while that patient also developed deafness at age 8 and DI at age 26 [30]. A second unrelated patient in the same study was reported to have developed DM, OA, and deafness, at the ages of 2, 20, and 6, respectively [30]. Similarly, a homozygous patient with the p.P320Sfs*39, which Patient 1 is heterozygous for, has also been described, and although no age of onset is provided, this individual had DM, OA, deafness, bladder dysfunction, and a psychiatric disorder [30]. A limitation of this study is the small cohort size, as a larger cohort would be ideal to further evaluate genotype–phenotype correlations and the various presentations of WS.

## 4. Conclusions

WS is a neurodegenerative disorder characterized by OA, DM, DI, and deafness. Most patients, however, do not develop the full phenotype, making WS a phenotypically diverse entity causing patients to have various clinical presentations. In this case series, we describe patients presenting for evaluation of acute visual decompensation superimposed on chronic vision loss, dyschromatopsia, and tonic pupils from a neuro-ophthalmology clinic at an academic medical center. Structural MRI has characteristic findings in WS and can be an important adjunct testing modality, while genetic testing is important to confirm the diagnosis. Genotype–phenotype correlations are an important tool for the clinician in determining severity and age of onset in clinical disorders, as missense mutations are typically associated with milder, later-onset WS. The presence of DM and/or OA early in life should prompt an appropriate work-up that includes screening for WS, as early diagnosis is crucial for the effective referrals for and management of the associated conditions by the appropriate providers. The condition should also be considered in later-onset, otherwise-unexplained, findings of the classic phenotype.

## Figures and Tables

**Figure 1 brainsci-13-01030-f001:**
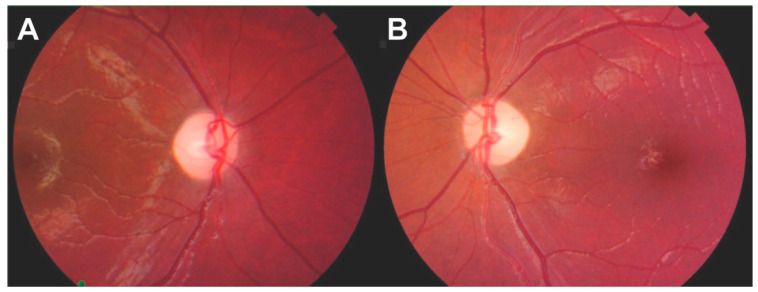
Fundoscopic examination of Patient 1. Fundoscopy revealed a diffusely pale optic disc with sharp borders in both the right (**A**) and left eye (**B**).

**Figure 2 brainsci-13-01030-f002:**
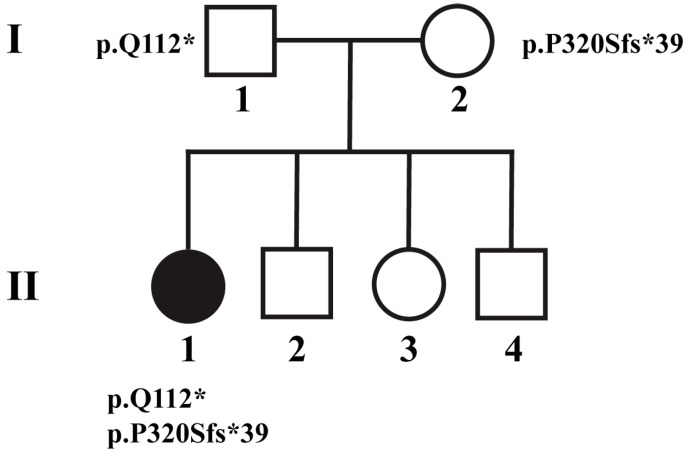
Pedigree of Patient 1. Patient 1 (II-1) is the only known affected sibling in a family of four, inheriting the p.Q112* variant from the father (**I**-**1**) and p.P320Sfs*39 from the mother (**I-2**). Sibling genotypes are unknown.

**Figure 3 brainsci-13-01030-f003:**
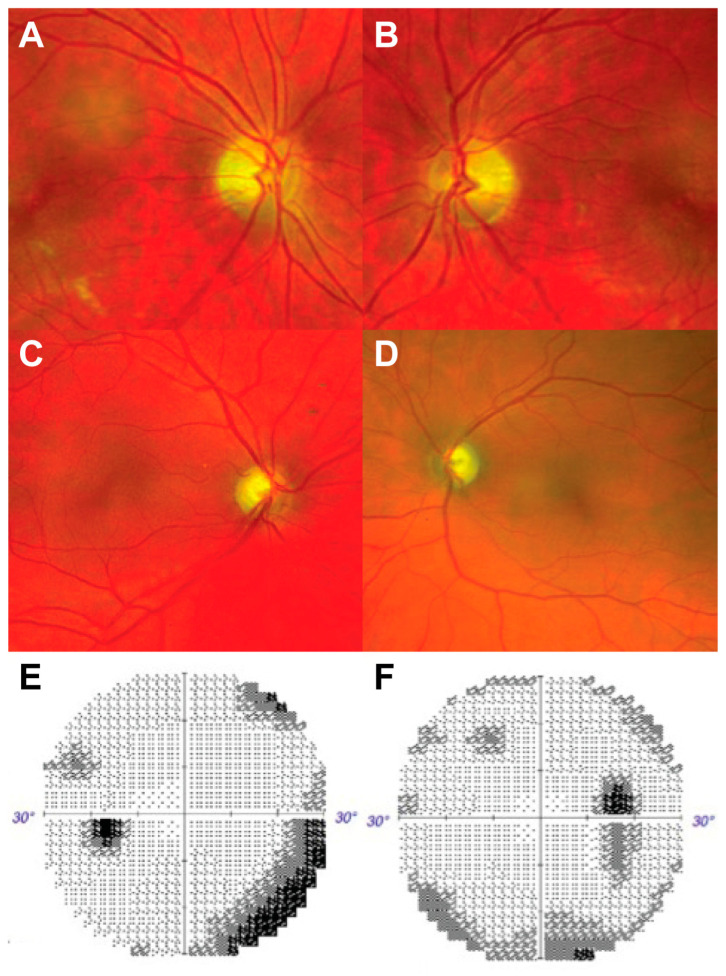
Fundoscopic examination and visual field testing of Patients 2 and 3. Fundoscopy on Patient 2 revealed pale optic nerves with sharp borders bilaterally, with thinning more pronounced temporally (**A**,**B**). Similar findings were observed in Patient 3 (**C**,**D**). Additionally, visual field testing in Patient 3 revealed an enlarged blind spot on the right eye (**F**) and an inferior nasal defect on the left eye (**E**).

**Figure 4 brainsci-13-01030-f004:**
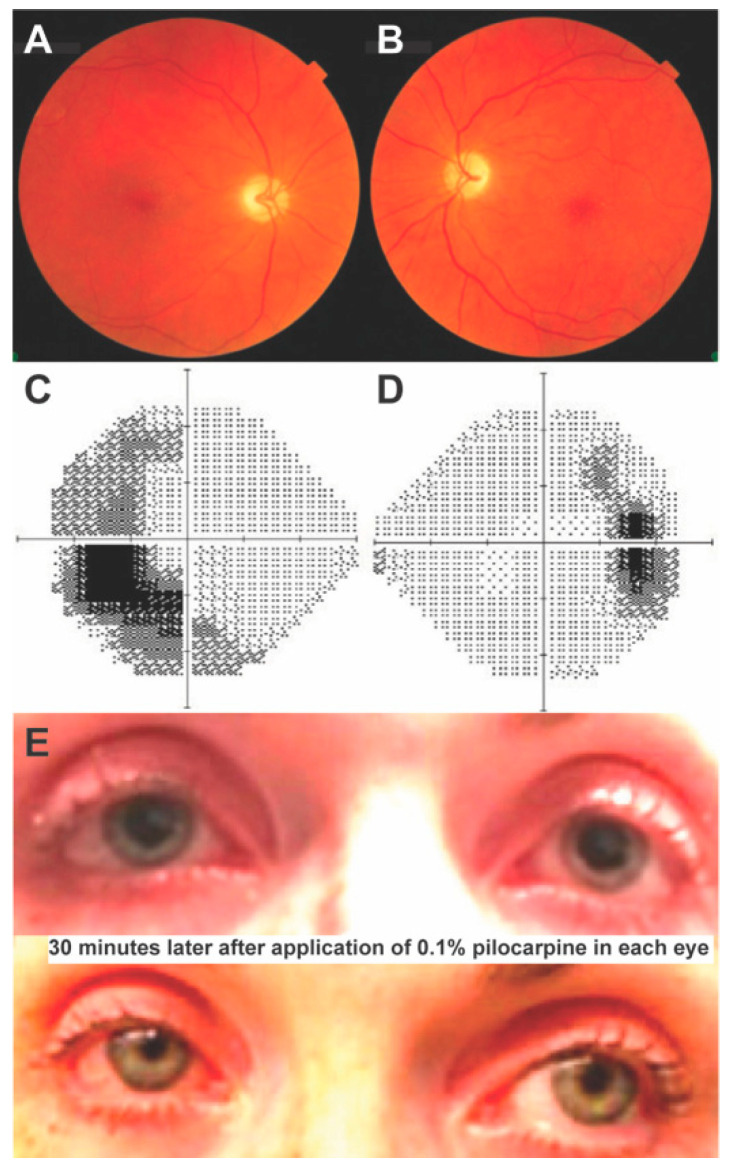
Fundoscopic examination, visual field testing, and pilocarpine test of Patient 4. Fundoscopy on Patient 4 revealed pale optic nerves bilaterally (**A**,**B**). Visual field testing revealed an enlarged blind spot on the right (**D**) and temporal field defects on the left eye (**C**). Pupils were dilated and unreactive to light, but after instilling dilute pilocarpine (0.1%) in both eyes, the pupils constricted, confirming a diagnosis of tonic pupils (**E**).

**Figure 5 brainsci-13-01030-f005:**
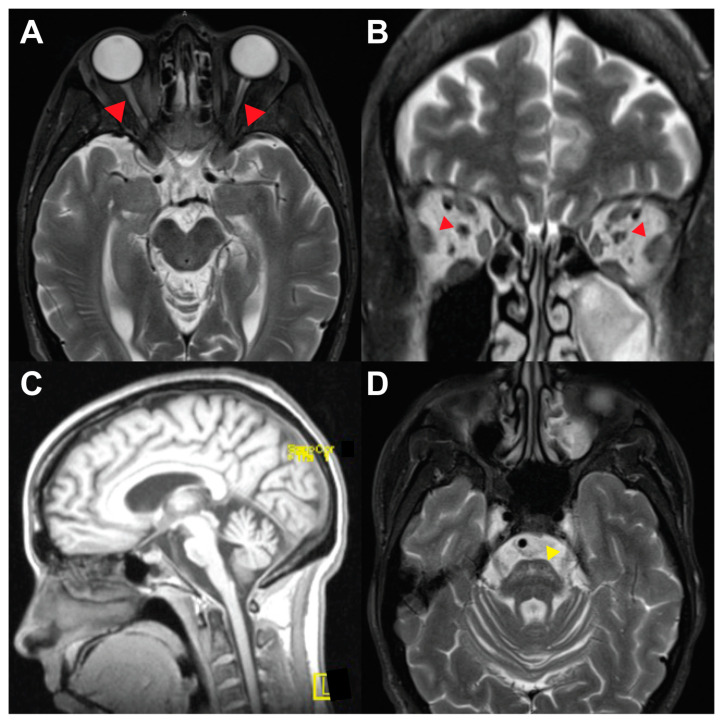
MRI brain and orbits for Patient 4. MRI revealed bilateral optic nerve atrophy, as appreciated on T2 axial (**A**) and coronal (**B**) sequences (red arrowheads). On T1 sagittal sequence, diffuse atrophy of the brainstem and cerebellum is appreciated (**C**), while hyper intensities in the pons (yellow arrowhead) are seen on T2 axial sequences (**D**).

**Table 1 brainsci-13-01030-t001:** Clinical summary of the patient cohort.

No.	Eval. Age	Sex	*WFS1* Variants	Presenting Sign/Symptom	DM	OA	DI	D	Other	MRI Findings
					(Age in Years at Onset)					
1	6	F	c.334 C>T:p.Q112*	Progressive vision loss	4	5	-	-	None	Unrevealing
			c.958_961delinsTCC:p.P320Sfs*39							
2	32	M	c.1672C>T:p.R558C	Color vision changes	20s	20s	-	-	None	Optic nerve atrophy
			c.1672C>T:p.R558C							
3	46	F	c.2254G>T:p.E752*	Progressive vision loss	-	Childhood	-	-	None	Optic nerve atrophy
			c.1673G>A:p.R558H							
4	45	F	c.1230_1233del:p.V412Sfs*29	Pupillary dilation	9	20s	-	-	Neurogenic bladder	Optic nerve, brainstem, cerebellum atrophy
			c.1672C>T:p.R558C	Progressive vision loss						Pons T2 hyper intensities

- = Not reported; D = deafness; DI = diabetes insipidus; DM = diabetes mellitus; Eval. Age = age at evaluation in years; No. = patient number; MRI = magnetic resonance imaging; OA = optic atrophy.

## Data Availability

The data presented in this study are available upon reasonable request from the corresponding author. The data are not publicly available to protect the privacy of our patients.

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
