# Peer review of "Neuro-Ophthalmologic Variability in Presentation of Genetically Confirmed Wolfram Syndrome: A Case Series and Review"

_brainsci, 2023, doi:10.3390/brainsci13071030_

Round 1

Reviewer 1 Report

The manuscript entitled "Neuro-ophthalmologic Variability in Presentation of Genetically-Confirmed Wolfram Syndrome: A Case Series and Review" can be accepted for publication. 

This can be acceptable

Author Response

Thank you for your remarks that this publication can be accepted.

Reviewer 2 Report

The paper entitled “Neuro-ophthalmologic Variability in Presentation of Genetically-Confirmed Wolfram Syndrome: A Case Series and Review” is based on a few patients with genetically confirmed Wolfram syndrome. The paper is quite interesting considering the variable clinical presentations (i.e. acute-on-chronic vision loss, dyschromatopsia, tonic pupils, optic atrophy, etc.).

The review shows that although this condition is rare, it should be considered in the differential diagnosis when there are important ophthalmologic and systemic clinical manifestations.  The paper is thorough and highlights the important issues behind Wolfram Syndrome. The study adds to the literature, especially considering the extremely low prevalence of this condition. The use of headings and subheadings gives the paper structure and logical organization.

Each case report is clearly presented, with appropriate clinical data and figures.  The study has been correctly planned and represents a solid basis for future studies regarding diagnosis and potential treatment options. It is nicely written and of clinical interest. References are appropriate.

The authors should add more information about all the possible pathologies and genetic disorders that can give similar signs and symptoms in the Discussion section. Foster Kennedy syndrome, for example, should be considered in the differential diagnosis as a possible cause of unilateral optic atrophy, with appropriate reference citations (i.e. Clin Pract. 2022 Jul 12;12(4):527-532. doi: 10.3390/clinpract12040056. PMID: 35892442).

The cohort is relatively small and thus should be reported as a limitation in the Discussion. A flowchart of signs, symptoms, clinical examinations, differential diagnosis, and therapies depending on individual findings could enhance the use in a routine clinical setting when managing patients with severe and variable ophthalmologic sight-threatening conditions such as the cases reported in this manuscript.

Minor editing of the English language can enhance the flow of the paper.

Author Response

1) Thank you.  We added a paragraph in the discussion about the differential diagnosis.  However, Foster Kennedy Syndrome may be less relevant to this discussion given that findings in FKS are typically asymmetric (monocular papilledema with concurrent pallor in the fellow eye) where as we would expect Wolfram fundoscopic exam to show symmetric findings.   

2) We have added a statement that asmall cohort is a limitation of this study. 

Reviewer 3 Report

My suggestions:

1. I would explain the structure and function of the WFS gene a little bit more in detail. 

2. I would add a table, which summarizes all cases, including clinical symptoms, mutations age of onset.

3. Was MRI done for the other patients too? If yes, was there any difference in their imaging data?

4. I would add a figure of the family tree for all patients, especially for patients with a positive family history of the disease.

5. Besides R558C, are the other mutations novel? If not, it would be nice to discuss the previously described cases in detail. 

Author Response

1) Thank you – this was addressed in the introduction with some new wording.

2) Thank you for this helpful suggestion, a table was added summarizing the cases. 

3) The discussion has reviewed the  MRI findings for all patients.

4) Thank you – we did add a figure but given limitations in scope of article we felt that adding a family tree would have been lower yield than a table, and we did not have complete information for all patients in this realm.   

5) Addressed in the discussion. We report 4 different variants in our study, all have been reported in literature, but for two there is not much info about the patients (it’s just reported in studies with many patients, but not much info for each individual patient). We elaborated on the other variants for which there is information in the current literature.

Reviewer 4 Report

Interesting syndrom and cases. For readers is very interesting.need iprove the introduction and discussion

Author Response

Thank you for your comments.  We improved the introduction and discussion with more information.

Round 2

Reviewer 3 Report

The authors fulfilled my suggestions, thank you